# The association between use of proton-pump inhibitors and excess mortality after kidney transplantation: A cohort study

Rianne M. Douwes[1☺]*, António W. Gomes-Neto[1☺], Michele F. Eisenga[1], Elisabet Van Loon[2], Joëlle C. Schutten[1], Rijk O. B. Gans[1], Maarten Naesens[2], Else van den Berg[1], Ben Sprangers[2], Stefan P. Berger[1], Gerjan Navis[1], Hans Blokzijl[3], Björn Meijers[2], Stephan J. L. Bakker[1], Dirk Kuypers[2]

1 Department of Internal Medicine, Division of Nephrology, University Medical Center Groningen, University of Groningen, Groningen, The Netherlands, 2 Department of Nephrology and Renal Transplantation, University Hospitals Leuven and Nephrology & Renal Transplantation Research Group, Department of Microbiology, Immunology and Transplantation, KU Leuven, Leuven, Belgium, 3 Department of Gastroenterology and Hepatology, University Medical Center Groningen, University of Groningen, Groningen, The Netherlands

☺ These authors contributed equally to this work.
* r.m.douwes@umcg.nl

**Data Availability Statement:** The data underlying the results presented in this study can be made available by the data manager of the

## Abstract

### Background

Chronic use of proton-pump inhibitors (PPIs) is common in kidney transplant recipients (KTRs). However, concerns are emerging about the potential long-term complications of PPI therapy. We aimed to investigate whether PPI use is associated with excess mortality risk in KTRs.

### Methods and findings

We investigated the association of PPI use with mortality risk using multivariable Cox proportional hazard regression analyses in a single-center prospective cohort of 703 stable outpatient KTRs, who visited the outpatient clinic of the University Medical Center Groningen (UMCG) between November 2008 and March 2011 (ClinicalTrials.gov Identifier NCT02811835). Independent replication of the results was performed in a prospective cohort of 656 KTRs from the University Hospitals Leuven (NCT01331668). Mean age was 53 ± 13 years, 57% were male, and 56.6% used PPIs. During median follow-up of 8.2 (4.7–9.0) years, 194 KTRs died. In univariable Cox regression analyses, PPI use was associated with an almost 2 times higher mortality risk (hazard ratio [HR] 1.86, 95% CI 1.38–2.52, $P <$ 0.001) compared with no use. After adjustment for potential confounders, PPI use remained independently associated with mortality (HR 1.68, 95% CI 1.21–2.33, $P =$ 0.002). Moreover, the HR for mortality risk in KTRs taking a high PPI dose (>20 mg omeprazole equivalents/ day) compared with patients taking no PPIs (HR 2.14, 95% CI 1.48–3.09, $P <$ 0.001) was higher than in KTRs taking a low PPI dose (HR 1.72, 95% CI 1.23–2.39, $P =$ 0.001). These findings were replicated in the Leuven Renal Transplant Cohort. The main limitation of this study is its observational design, which precludes conclusions about causation.

TransplantLines study, by mailing to transplantlines@umcg.nl. Data from the BIOBANK Renal Transplantation University Hospitals Leuven can be made available by mailing the ethics committee of the University Hospitals Leuven EC@UZLEUVEN.BE. Public sharing of individual participant data was not included in the informed consent forms of both studies, but data can be made available to interested researchers upon reasonable request.

**Funding:** Generation of this study was funded by Top Institute Food and Nutrition. RMD is supported by NWO/TTW in a partnership program with DSM, Animal Nutrition and Health, The Netherlands; project number: 14939. EVL holds a fellowship grant (1143919N) from the Research Foundation Flanders (F.W.O.). The funders had no role in study design, data collection and analysis, decision to publish, or preparation of the manuscript.

**Competing interests:** The authors have declared that no competing interests exist.

**Abbreviations:** BMI, body mass index; BPAR, biopsy proven acute rejection; CKD-EPI, Chronic Kidney Disease Epidemiology Collaboration; CNI, calcineurin inhibitor; eGFR, estimated glomerular filtration rate; FDA, Food and Drug Administration; HbA1c, hemoglobin A1c; HDL, high-density lipoprotein; HLA, Human Leukocyte Antigen; HR, hazard ratio; ICD-9, International Classification of Diseases, Ninth Revision; ICD-10, International Classification of Diseases, Tenth Revision; IQR, interquartile range; KTR, kidney transplant recipient; LDL, low-density lipoprotein; NSAID, nonsteroidal anti-inflammatory drug; PPI, proton-pump inhibitor; STROBE, Strengthening the Reporting of Observational Studies in Epidemiology; TRIPOD, Transparent Reporting of a Multivariable Prediction Model for Individual Prognosis or Diagnosis; UMCG, University Medical Center Groningen.

## Conclusions

We demonstrated that PPI use is associated with an increased mortality risk in KTRs, independent of potential confounders. Moreover, our data suggest that this risk is highest among KTRs taking high PPI dosages. Because of the observational nature of our data, our results require further corroboration before it can be recommended to avoid the long-term use of PPIs in KTRs.

## Trial registration

ClinicalTrials.gov Identifier: NCT02811835, NCT01331668.

## Author summary

### Why was this study done?

- Proton-pump inhibitors (PPIs) are commonly prescribed to prevent gastrointestinal side effects of immunosuppressive medication after kidney transplantation, and there is little incentive to discontinue use of PPIs in the long term.

- Several observational studies among individuals from the general population and among patients on hemodialysis have found that PPI use is associated with a higher mortality risk.

- Long-term mortality rates in kidney transplant recipients (KTRs) are high. Therefore, we aimed to investigate whether PPI use is associated with increased mortality risk in KTRs.

### What did the researchers do and find?

- We performed a post hoc analysis using data from the TransplantLines Food and Nutrition Biobank and Cohort Study, a prospective cohort study in 703 KTRs, with baseline assessments performed between November 2008 and March 2011. Follow-up was performed for a median of 8.2 years.

- We found that PPI users had an almost 2-fold increased mortality risk compared with nonusers. When we looked at the cause of death, we found that PPI use was particularly associated with mortality due to cardiovascular diseases and infectious diseases. We also demonstrated that mortality risk is highest among KTRs taking high PPI dosages (>20 mg omeprazole equivalents/day).

- These findings were replicated in an independent cohort of 656 KTRs from the University Hospitals Leuven, which strengthens the evidence for an association between PPI use and mortality risk in KTRs.

**What do these findings mean?**

- Results of this study suggest that PPI use is associated with mortality risk in KTRs, independent of potential confounders.

- The current study highlights the importance of an evidence-based indication for PPI treatment and provides a rationale to perform a randomized controlled trial on chronic PPI therapy in KTRs.

## 1. Introduction

Renal transplantation is considered the preferred treatment for patients with end-stage renal disease, providing improved prognosis and quality of life at lower cost compared with dialysis treatment [1–3]. Although short-term outcomes after renal transplantation have tremendously improved over the last decades, long-term graft survival and mortality rates have shown little improvement [4–6]. Indeed, mortality rates in kidney transplant recipients (KTRs) are still 6 times higher than in the general population [7].

In search of modifiable risk factors of this excess mortality, iatrogenic factors should not be overlooked. In this respect, proton-pump inhibitors (PPIs) have drawn our attention. PPIs are commonly prescribed to KTRs to prevent dyspeptic symptoms and complications from immunosuppressive agents. Despite the favorable safety profile of these drugs, which are generally well tolerated, growing concern and evidence about the potential long-term complications of PPI therapy are emerging. Since the first case report of PPI induced hypomagnesemia [8], numerous observational studies have demonstrated that chronic PPI use is associated with several adverse health outcomes, including increased risk of nutrient deficiencies [9–15], *Clostridium difficile* infections [16,17], community acquired pneumonia [18], acute and chronic kidney disease [19–21], and end-stage renal disease [22]. Given that KTRs are prone to nutrient deficiencies, have a high burden of premature cardiovascular morbidity, and recurrent infections due to use of immunosuppressive medication, KTRs might especially be susceptible to adverse effects of PPI use.

Recently, several observational studies have demonstrated that PPI use may be associated with an increased risk of mortality in elderly patients [23–25]. Interestingly, the same prospective association between PPI use and increased mortality risk was found in a large cohort study of United States veterans [26] and in a cohort of 1,776 hemodialysis patients [27]. Whether chronic PPI use is associated with an increased risk of mortality in KTRs is currently unknown. Therefore, we investigated the effect of PPI therapy on mortality in a large single-center cohort of stable outpatient KTRs. Because of the observational nature of our data—and the fact that, for the primary cohort, baseline assessments were performed at varying time after transplantation, which could have induced survival bias—we investigated whether our findings could be replicated in an independent cohort of stable KTRs, in which baseline assessments have been conducted without variation in time after transplantation [28].

## 2. Methods

### 2.1 Design and study population

This is a post hoc analysis using data from a previously described cohort of 707 stable KTRs, registered as the TransplantLines Food and Nutrition Biobank and Cohort Study

(ClinicalTrials.gov Identifier NCT02811835), which is a prospective cohort study intended to investigate the relationship between dietary acid load, ammoniagenesis, and its potential influence on blood pressure [29]. In summary, all adult KTRs with a functioning graft for at least 1 year after transplantation who visited the outpatient clinic of the University Medical Center Groningen (UMCG) between November 2008 and March 2011 were invited to participate in the study. KTRs were not considered eligible for the study in case of concurrent systemic illnesses, including malignancies other than cured skin cancer, opportunistic infections, and overt congestive heart failure. Of the initially 817 invited KTRs, 707 (87%) gave written informed consent. We excluded KTRs with missing data on PPI dosage (n = 1) or with on-demand PPI use (n = 3), leaving 703 KTRs eligible for the current post hoc analysis. All measurements were performed during a single study visit at the outpatient clinic. The primary endpoint of this study was all-cause mortality. In response to peer-review comments, we added cause-specific mortality (i.e., death due to cardiovascular diseases, infectious diseases, malignant diseases, and miscellaneous causes) and occurrence of graft failure as secondary endpoints. Follow-up was recorded until September 2015, and upon request of one of the reviewers, it was extended to December 31, 2018. Continuous surveillance of the outpatient program ensures up-to-date information on patient status, which was recorded in the UMCG Renal Transplantation Database and verified with the Dutch Civil Registration Office. Medical records, general practitioners, and nephrologist were consulted to establish cause of death. Cardiovascular mortality was defined as death due to cerebrovascular disease, ischemic heart disease, heart failure, or sudden cardiac death International Classification of Diseases, Ninth Revision (ICD-9) codes 410–447, infectious disease mortality was defined according to ICD-9 codes 1–139, and cancer mortality was defined according to a specified list of ICD-9 codes [30]. The study protocol was approved by the institutional review board of the UMCG (IRB identifier 2008–186). All study procedures were performed in accordance with the Declaration of Helsinki and the Declaration of Istanbul.

For the replication study, we used data from an independent cohort of stable KTRs from the University Hospitals Leuven [28]. In summary, in the University Hospitals Leuven Renal Transplant Program, the majority of patients (>95%) are enrolled in a prospective Renal Transplant Biobank Program (ClinicalTrials.gov identifier NCT01331668). We used data from a previously described cohort in which patients were seen at the outpatient clinic at 3, 12, and 24 months after transplantation, and yearly thereafter [28]. During the outpatient clinic visit, routine laboratory analyses were performed together with a physical examination. Survival time was defined from the date of the last study visit until date of death or end of follow-up. KTRs who developed graft failure during follow-up (i.e., return to dialysis or re-transplantation) were censored at time of graft failure. Information on clinical parameters, including medication use, weight, and laboratory results, was obtained from electronic clinical patient charts. All participants provided written informed consent. The study was approved by the Ethics Committee of the University Hospitals Leuven (S53364; ML7499). This study is reported as per the Strengthening the Reporting of Observational Studies in Epidemiology (STROBE) guideline (S1 STROBE Checklist) and Transparent Reporting of a Multivariable Prediction Model for Individual Prognosis or Diagnosis (TRIPOD) guideline (S1 TRIPOD Checklist).

## 2.2 Clinical parameters and measurements

Information on medical history, including primary renal disease, was obtained from patient records [29]. Transplant-specific characteristics were retrieved from the local University Medical Center Groningen Renal Transplantation Database. History of cardiovascular disease was

classified according to International Classification of Diseases, Tenth Revision (ICD-10) codes Z86.7. KTRs were considered to have diabetes when at least one of the following criteria was met: (1) symptoms of diabetes (e.g., polyuria, polydipsia, unexplained weight loss) plus casual plasma glucose concentration of $\geq$11.1 mmoL/L (200 mg/dL), (2) fasting plasma glucose concentration $\geq$7.0 mmol/L (126 mg/dL), (3) use of antidiabetic medication, or (4) plasma hemoglobin A1c (HbA1c) $\geq$6.5% (48 mmol/L). Body mass index (BMI) was calculated as weight in kilograms divided by height in meters squared. Blood pressure was measured according to a strict protocol, as previously described in detail [31]. Information on alcohol use and smoking behavior was obtained using a questionnaire (see S1 Appendix). Medication use, including use of PPIs, was recorded at baseline and verified with medical records. KTRs using any PPI on a daily basis during a period of at least 3 months before and 3 months after the study visit were defined as chronic PPI users.

Blood samples were collected after an 8- to 12-hour overnight fasting period. Serum creatinine was measured using an enzymatic, isotope dilution mass spectrometry–traceable assay (P-Modular automated analyzer, Roche Diagnostics, Basel, Switzerland). Estimated glomerular filtration rate (eGFR) was calculated applying the serum creatinine–based Chronic Kidney Disease Epidemiology Collaboration (CKD-EPI) equation [32]. Concentrations of cholesterol, triglycerides, and HbA1c were determined using standard laboratory methods. All participants were instructed to collect a 24-hour urine sample according to a strict protocol on the day prior to their visit to the outpatient clinic. Urine was collected under oil, and chlorhexidine was added as antiseptic agent. Total urinary protein concentration was determined using the Biuret reaction (MEGA AU 510, Merck Diagnostica, Darmstadt, Germany). Proteinuria was defined as urinary protein excretion $\geq$0.5 g/24 h.

## 2.3 Statistical analyses

Statistical analyses were performed with SPSS software, version 23.0 (IBM Corp., Armonk, NY) and Stata version 14.2 (StataCorp LP, College Station, TX). Data are presented as mean ± SD for normally distributed data or as median with interquartile range (IQR) for non-normally distributed data and number with percentage for nominal data. Differences between PPI users and nonusers were tested using an independent-sample $t$ test for normally distributed data, Mann-Whitney U test for non-normally distributed data, or chi-squared test for categorical data.

A Kaplan-Meier curve was used to illustrate the association of PPI use on patient survival, and significance was tested using the log-rank test. Survival time was defined as the time from baseline visit until the date of death or end of follow-up (December 31, 2018). KTRs who developed graft failure during follow-up (i.e., return to dialysis or re-transplantation) were censored at time of graft failure. Multivariable Cox proportional hazard regression analyses were performed to analyze whether the hypothesized association of PPI use with mortality was independent of potential confounders. To adjust for confounders, multiple models were built. In model 1, we adjusted for age, sex, BMI, and time since transplantation; in model 2, we further adjusted for eGFR and proteinuria, deceased donor transplant, preemptive transplantation, and primary renal disease, cumulative to already existing adjustments performed in model 1. Because of the limited number of events and the rule of thumb of allowing for one covariate per approximately 10 events to prevent overfitting and overadjustment of the models, further models were performed with additive adjustments to model 2 [33,34]. In model 3, we additionally adjusted for donor characteristics and immunological factors (donor age, donor sex, donor height, donor weight, donor serum creatinine, number of Human Leukocyte Antigen [HLA] mismatches, and applied induction therapy). (This model was added in

response to a comment from one of the reviewers.) In model 4, we adjusted for lifestyle parameters (alcohol use and smoking behavior). In model 5, we adjusted for medication use (antihypertensive drugs, platelet inhibitors, vitamin K antagonists, proliferation inhibitors, and calcineurin inhibitors [CNIs]), and in model 6, we adjusted for comorbidities (diabetes and history of cardiovascular disease), in addition to adjustments in model 2. In model 7, we performed adjustment for plasma magnesium and serum iron, in addition to adjustments in model 2, to investigate whether these variables were potential mediators of the association between PPI use and mortality. In response to a request from one of the reviewers, we additionally performed analyses in which we adjusted for all covariates from models 1–6 in a final cumulative model. To avoid overadjustment bias due to inclusion of overlapping covariates within one biological domain, we ran 2 analyses. In one analysis we excluded diabetic nephropathy, vitamin K antagonists, platelet inhibitors, and use of anti-hypertensive drugs from the final model, and in another analysis we excluded diabetes and cardiovascular disease history from the final model [35,36]. The proportional hazards assumption was tested using the Schoenfeld global test and was not violated in any of the models. In response to peer-review comments, we added analyses on potential effect modification by age, sex, eGFR, diabetes, and medication for which significant baseline differences were present, including the use of antihypertensive drugs, platelet inhibitors, vitamin K antagonists, and CNIs. This was tested by adding interaction terms consisting of PPI use and the variable of interest to model 2 of the Cox regression analyses.

To explore a potential dose-response relationship, we performed additional Cox regression analyses in which KTRs were divided into 3 groups based on daily PPI dose defined in omeprazole equivalents: no PPI, low PPI dose ($\leq$20 mg omeprazole equivalents/day), and high PPI dose (>20 mg omeprazole equivalents/day) as described previously [37]. Tests of linear trend were conducted by assigning the median of daily PPI dose equivalents in subgroups treated as a continuous variable.

In response to peer-review comments, we performed secondary analyses for the association of PPI use with cause-specific mortality (i.e., death due to cardiovascular diseases, infectious diseases, malignant diseases, and miscellaneous causes), death-censored graft failure, and biopsy proven acute rejection (BPAR). Because of lower numbers of events available for these endpoints, analyses were limited to the first 2 models. We also investigated whether PPI use was associated with eGFR decline, because PPI use has been associated with acute interstitial nephritis in the past. Information on the last measurement of serum creatinine before occurrence of either death, death-censored graft failure, or end of follow-up was obtained from medical records. These levels were used to calculate the eGFR at the moment closest to patient death. The delta eGFR in mL/min/1.73 m$^2$ was calculated by subtracting the baseline eGFR from the eGFR at follow-up. Linear regression analyses with delta eGFR as dependent variable and PPI use as independent variable were performed to investigate the association of PPI use with change in eGFR. Furthermore, in response to peer-review comments, we performed sensitivity analyses in which we adjusted for eGFR change cumulative to already existing adjustments performed in model 2 of the Cox regression analyses. We performed multiple imputations ($n$ = 5) to account for missing data on baseline characteristics in our Cox regression analyses [38]. Results of the imputed dataset were compared with the results of the non-imputed dataset and showed no relevant differences (S1 Table). In all analyses, a two-sided $P < 0.05$ was considered statistically significant.

## 3. Results

Baseline characteristics of the TransplantLines study are shown in Table 1. Mean age at baseline was 53 ± 13 years, and 401 (57.0%) KTRs were male. Mean BMI was 26.7 ± 4.8 kg/m$^2$, and

**Table 1. Baseline characteristics of 703 KTRs from the TransplantLines study.**

| Characteristics | | | Total population | Non-PPI users | PPI users | P |
|---|---|---|---|---|---|---|
| Number of participants , n (%) | | | 703(100) | 305 (43.4) | 398 (56.6) | n/a |
| Demographics | | | | | | |
| | Age, y | | 53 ± 13 | 51 ± 13 | 54 ± 12 | 0.001 |
| | Men, n (%) | | 401 (57.0) | 179 (58.7) | 222 (55.8) | 0.4 |
| | Height, cm | | 174 ± 10 | 174 ± 10 | 174 ± 10 | 1.0 |
| | BMI, kg/m$^2$ | | 26.7 ± 4.8 | 26.0 ± 4.6 | 27.2 ± 4.9 | 0.001 |
| | Diabetes mellitus, n (%) | | 168 (23.9) | 56 (18.4) | 112 (28.1) | 0.003 |
| | Cardiovascular disease history, n (%) | | 280 (39.8) | 95 (31.1) | 185 (46.5) | <0.001 |
| Lifestyle parameters | | | | | | |
| | Current smoker, n (%)[a] | | 84 (12.8) | 35 (12.2) | 49 (13.3) | 0.7 |
| | Alcohol consumer, n (%)[a] | | 442 (69.8) | 199 (72.1) | 243 (68.1) | 0.3 |
| Primary renal disease | | | | | | |
| | | Glomerulonephritis, n (%) | 189 (26.9) | 89 (29.2) | 100 (25.1) | 0.2 |
| | | Interstitial nephritis, n (%) | 86 (12.2) | 49 (16.1) | 37 (9.3) | 0.007 |
| | | Cystic kidney disease, n (%) | 145 (20.6) | 53 (17.4) | 92 (23.1) | 0.06 |
| | | Other congenital and hereditary kidney disease, n (%) | 42 (6.0) | 24 (7.9) | 18 (4.5) | 0.06 |
| | | Renal vascular disease, n (%) | 38 (5.4) | 19 (6.2) | 19 (4.8) | 0.4 |
| | | Diabetes Mellitus, n (%) | 36 (5.1) | 8 (2.6) | 28 (7.0) | 0.009 |
| | | Other multisystem diseases, n (%) | 49 (7.0) | 17 (5.6) | 32 (8.0) | 0.2 |
| | | Other, n (%) | 16 (2.3) | 7 (2.3) | 9 (2.3) | 1.0 |
| | | Unknown, n (%) | 102 (14.5) | 39 (12.8) | 63 (15.8) | 0.3 |
| Transplantation characteristics | | | | | | |
| | Time since transplantation, y | | 5.4 [1.9-12.0] | 9.4 [4.1-15.0] | 4.1 [1.2-8.5] | <0.001 |
| | Pre-emptive transplantation, n (%) | | 113 (16.1) | 45 (14.8) | 68 (17.1) | 0.4 |
| | Total HLA mismatches [a] | | 2 [1 – 3] | 2 [1 – 3] | 2 [1 – 3] | 0.04 |
| | Induction therapy [b] | | | | | <0.001 |
| | | Anti-thymocyte globulin | 61 (9.1) | 20 (7.1) | 41 (10.6) | |
| | | CD3 receptor MoAb | 16 (2.4) | 11 (3.9) | 5 (1.3) | |
| | | IL2 receptor MoAb | 348 (52.0) | 106 (37.7) | 242 (62.4) | |
| | | Other | 29 (4.3) | 15 (5.3) | 14 (3.6) | |
| | | None | 215 (32.1) | 129 (45.9) | 86 (22.2) | |
| Donor characteristics | | | | | | |
| | Deceased donor, n (%) | | 462 (65.9) | 212 (69.7) | 250 (63.0) | 0.06 |
| | Men, n (%)[b] | | 355 (51.7) | 156 (53.2) | 199 (50.5) | 0.5 |
| | Age, y [b] | | 42 ± 15 | 40 ± 16 | 45 ± 14 | <0.001 |
| | Weight, kg [c] | | 75.8 ± 15.4 | 74.8 ± 13.3 | 76.6 ± 16.6 | 0.2 |
| | Height, cm [c] | | 174 ± 11 | 175 ± 9 | 174 ± 12 | 0.7 |
| | Creatinine, μmol/L [c] | | 75 [62 – 93] | 75 [62 – 93] | 80 [63 – 92] | 0.9 |
| Renal function parameters | | | | | | |
| | eGFR, ml/min/1.73 m$^2$ | | 52.2 ± 20.1 | 54.8 ± 19.9 | 50.1 ± 20.0 | 0.002 |
| | Serum creatinine, μmol/L | | 125 [100-161] | 119 [99-153] | 128 [101-168] | 0.04 |
| | Proteinuria (≥0.5 g/24h), n (%) | | 158 (22.5) | 71 (23.3) | 87 (22.0) | 0.7 |
| Hemodynamic parameters | | | | | | |
| | Systolic blood pressure, mmHg | | 136 ± 18 | 133 ± 17 | 138 ±18 | <0.001 |
| | Diastolic blood pressure, mmHg | | 83 ± 11 | 82 ± 11 | 83 ± 11 | 0.4 |
| | Heart rate, bpm[c] | | 69 ± 12 | 67 ± 12 | 70 ± 12 | 0.02 |
| Laboratory parameters | | | | | | |

(*Continued*)

**Table 1.** (Continued)

| Characteristics | Total population | Non-PPI users | PPI users | P |
|---|---|---|---|---|
| Magnesium, mmol/L | 0.77 ± 0.11 | 0.79 ± 0.09 | 0.76 ± 0.12 | <0.001 |
| Iron, μmol/L | 15.3 ± 6.0 | 16.4 ± 6.1 | 14.4 ± 5.8 | <0.001 |
| Total cholesterol, mmol/L | 5.1 ± 1.1 | 5.2 ± 1.1 | 5.1 ± 1.2 | 0.5 |
| HDL-cholesterol, mmol/L[b] | 1.3 [1.1-1.6] | 1.4 [1.1-1.7] | 1.3 [1.0-1.6] | <0.001 |
| LDL-cholesterol, mmol/L[b] | 3.0 ± 0.9 | 3.0 ± 1.0 | 2.9 ± 0.9 | 0.2 |
| Triglycerides, mmol/L | 1.7 [1.3-2.3] | 1.6 [1.1-2.1] | 1.7 [1.3-2.5] | 0.002 |
| Glucose, mmol/L | 5.3 [4.8-6.0] | 5.2 [4.7-5.8] | 5.3 [4.8-6.2] | 0.02 |
| HbA1c, %[b] | 5.8 [5.5-6.2] | 5.7 [5.4-6.0] | 5.9 [5.6-6.3] | <0.001 |
| Medication use | | | | |
| Antihypertensive drugs, n (%) | 620 (88.2) | 252 (82.6) | 368 (92.5) | <0.001 |
| Platelet inhibitors, n (%) | 144 (20.5) | 48 (15.7) | 96 (24.1) | 0.006 |
| Dual antiplatelet therapy, n (%) | 7 (1) | 1 (0.3) | 6 (1.5) | 0.2 |
| Vitamin K antagonists, n (%) | 77 (11.0) | 21 (6.9) | 56 (14.1) | 0.003 |
| Statins, n (%) | 371 (52.8) | 148 (48.5) | 223 (56.0) | 0.05 |
| Proliferation inhibitors, n (%) | 583 (82.9) | 251 (82.3) | 332 (83.4) | 0.7 |
| CNIs, n (%) | 406 (57.8) | 150 (49.2) | 256 (64.3) | <0.001 |
| Prednisolone, n (%) | 696 (99.0) | 303 (99.3) | 393 (98.7) | 0.7 |

Data are presented as mean ± SD, median with IQRs, or number with percentages (%).

[a]Missing in <10%.

[b]Missing in <5%.

[c]Missing in <20%.

**Abbreviations:** BMI, body mass index; CD3 MoAb; CD3 monoclonal antibody; CNI, calcineurin inhibitor; eGFR, estimated glomerular filtration rate; HbA1c, hemoglobin A1c; HDL, high-density lipoprotein; HLA, Human Leukocyte Antigen; IL2 MoAb, interleukin-2 receptor monoclonal antibody; IQR, interquartile range; KTR, kidney transplant recipient; LDL, low-density lipoprotein; n/a, not applicable; PPI, proton-pump inhibitor

168 (23.9%) KTRs met the criteria for diabetes. KTRs were included in the study at a median of 5.4 (1.9–12.0) years post transplantation, and 462 (65.9%) KTRs received a kidney transplant from a deceased donor. Mean eGFR was 52.2 ± 20.1 ml/min/1.73 m$^2$, and 158 (22.5%) KTRs had proteinuria. A small majority of 398 (56.6%) KTRs used PPIs. The most commonly used PPI was omeprazole ($n = 347$) accounting for 87% of all PPIs used, followed by esomeprazole ($n = 31$), pantoprazole ($n = 17$), and rabeprazole ($n = 3$). At baseline, we observed that PPI users were significantly older compared with nonusers, had a higher BMI, and were included with a shorter time interval between transplantation and baseline measurements. Furthermore, diabetes was more common among PPI users, and KTRs who used PPIs had higher systolic blood pressure, heart rate, and HbA1c levels and lower levels of high-density lipoprotein (HDL) cholesterol. Additionally, treatment with other medications, including antihypertensive dugs, platelet inhibitors, vitamin K antagonists, and CNIs, was more prevalent among PPI users compared with nonusers (Table 1). Of the KTRs using platelet inhibitors, 1 PPI nonuser (0.3%) versus 6 PPI users (1.5%) were on dual antiplatelet medication ($P = 0.2$).

### 3.1 PPI use and mortality risk

Median follow-up of the TransplantLines study was 8.2 years. During this period, 110 KTRs developed graft failure and were censored at time of graft failure, and 194 (27.6%) KTRs died with a functioning graft. The majority of KTRs died due to cardiovascular disease (37.1%), followed by death due to infectious diseases (24.2%), miscellaneous causes (20.1%), and

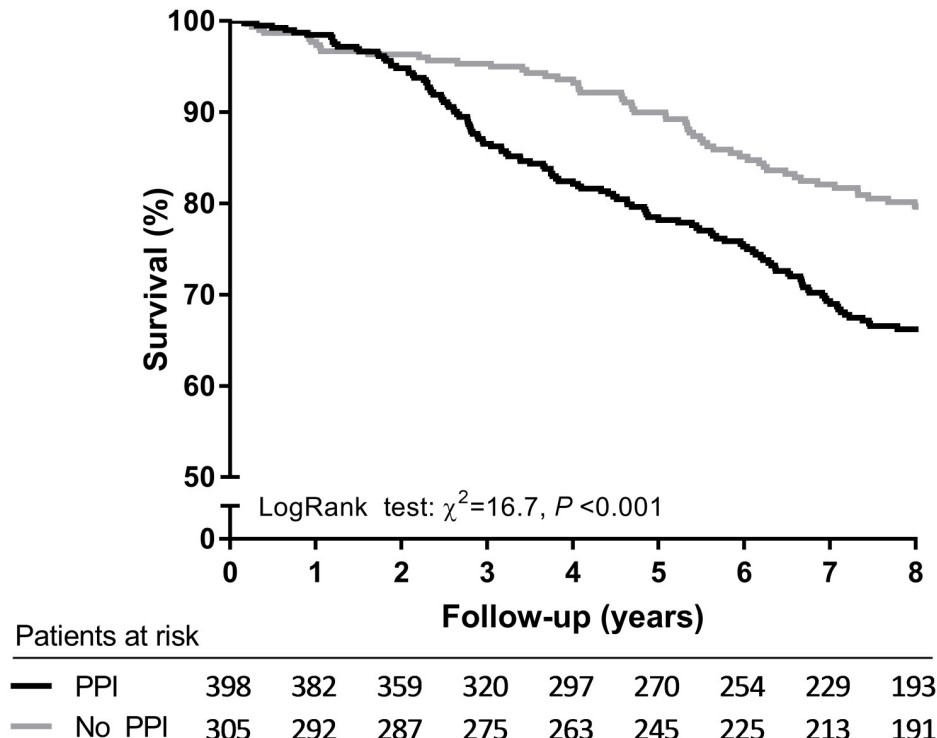

**Fig 1. Kaplan-Meier survival curve for all-cause mortality of PPI users compared with nonusers from the TransplantLines study.** PPI, proton-pump inhibitor.

malignant diseases (18.6%). Out of the 398 KTRs who used PPIs at baseline, 132 (33.2%) died during follow-up, whereas out of the 305 KTRs who did not use PPIs at baseline, 62 (20.3%) died during follow-up (log-rank test $P < 0.001$, Fig 1). Cox regression analysis revealed that PPI users had an increased risk of mortality compared with nonusers (hazard ratio [HR] 1.86, 95% CI 1.38–2.52, $P < 0.001$, Table 2). Adjustment for potential confounders—including age, sex, BMI, eGFR, proteinuria, time since transplantation, preemptive transplantation, deceased donor transplant, and primary renal disease—did not substantially affect the association (HR 1.68, 95% CI 1.21–2.33, $P = 0.002$, Table 2). Additionally, the association remained independent of adjustment for other potential confounding factors, including donor characteristics, immunological factors, lifestyle parameters, medication use, and comorbidities. Adjustment for plasma magnesium and serum iron as potential mediators of the association between PPI use and mortality did not materially alter the association (HR 1.53, 95% CI 1.09–2.14, $P = 0.01$, Table 2). In a final model in which we adjusted for all variables of models 1–6 combined (except diabetic nephropathy, vitamin K antagonists, platelet inhibitors, and use of antihypertensive drugs), the association between PPI use and all-cause mortality remained (HR 1.42, 95% CI 1.01–2.00, $P = 0.04$). Alternatively, the association between PPI use and all-cause mortality remained in a final model in which we adjusted for all variables of models 1–6 combined except diabetes and cardiovascular disease history (HR 1.46, 95% CI 1.03–2.05, $P = 0.03$). No significant interactions were found between PPI use and age, sex, eGFR, diabetes, and medication for which significant baseline differences were present—including antihypertensive drugs, platelet inhibitors, vitamin K antagonists, and CNIs ($P_{interaction} > 0.05$).

**Table 2. Association of PPI use with mortality in 703 stable KTRs from the TransplantLines study.**

| | All-Cause Mortality | | |
|---|---|---|---|
| Number of events = 194 | HR | 95% CI | *P* |
| Crude | 1.86 | 1.38–2.52 | <0.001 |
| Model 1 | 1.73 | 1.25–2.38 | 0.001 |
| Model 2 | 1.68 | 1.21–2.33 | 0.002 |
| Model 3 | 1.67 | 1.19–2.34 | 0.003 |
| Model 4 | 1.63 | 1.17–2.27 | 0.007 |
| Model 5 | 1.49 | 1.07–2.09 | 0.02 |
| Model 6 | 1.46 | 1.04–2.03 | 0.03 |
| Model 7 | 1.53 | 1.09–2.14 | 0.01 |

Model 1: PPI use adjusted for age, sex, BMI, and time since transplantation. Model 2: Model 1 additionally adjusted for eGFR, proteinuria, deceased donor transplant, preemptive transplantation, and primary renal disease. Model 3: Model 2 additionally adjusted for donor age, donor sex, donor weight, donor height, donor serum creatinine, number of HLA mismatches, and induction therapy. Model 4: Model 2 additionally adjusted for smoking behavior and alcohol use. Model 5: Model 2 additionally adjusted for the use of antihypertensive agents, platelet inhibitors, vitamin K antagonists, proliferation inhibitors, and CNIs. Model 6: Model 2 additionally adjusted for comorbidities (diabetes, history of cardiovascular disease). Model 7: Model 2 additionally adjusted for potential mediators (plasma magnesium and serum iron).

**Abbreviations:** BMI, body mass index; CNI, calcineurin inhibitor; eGFR, estimated glomerular filtration rate; HLA, Human Leukocyte Antigen; HR, hazard ratio; KTR, kidney transplant recipient; PPI, proton-pump inhibitor

Cause-specific analyses revealed that PPI use was particularly associated with an increased risk of cardiovascular mortality (HR 2.42, 95% CI 1.43–4.08, $P < 0.001$). This association remained significant after adjustment for potential confounding factors (S2 Table). Moreover, we found an increased mortality risk due to infectious diseases among PPI users (HR 1.89, 95% CI 1.02–3.49, $P = 0.04$), although the association was slightly attenuated after adjustment for potential confounders (HR 1.88, 95% CI 0.96–3.71, $P = 0.07$, S2 Table). We did not observe a significant association between PPI use and death due to malignant diseases or miscellaneous causes (S2 Table). Furthermore, we found that PPI use was not significantly associated with a higher risk of graft failure (HR 1.20, 95% CI 0.82–1.75, $P = 0.4$, S3 Table). Only 13 KTRs developed BPAR during follow-up. We found no significant association between PPI use and subsequent development of BPAR (HR 1.73, 95% CI 0.53–5.61, $P = 0.4$). Unfortunately, the low number of events does not allow for meaningful analyses with adjustment for potential confounders of this association.

Median time between baseline eGFR and eGFR at follow-up was 6.5 (4.6–8.6) years. Mean change in renal function over this period was $-7.13 \pm 17.1$ ml/min/1.73 m$^2$. In crude linear regression analysis, PPI use was not associated with eGFR decline ($\beta = 0.75$, 95% CI $-1.82$ to 3.32, $P = 0.6$, S4 Table). Additional adjustment for time from baseline to follow-up, age, sex, and BMI did not substantially alter the association ($\beta = 0.10$, 95% CI $-2.45$ to 2.66, $P = 0.9$, S4 Table). Results from Cox regression analyses for the association of PPI use with all-cause mortality analogous to model 2 remained materially unchanged when we adjusted for change in eGFR (HR 1.68, 95% CI 1.22–2.33, $P = 0.002$).

## Dose response analysis

We also investigated whether KTRs taking a high PPI dose (>20 mg omeprazole equivalents/day) were at a higher risk for premature mortality compared with KTRs on a low PPI dose (≤20 mg omeprazole equivalents/day). At baseline, 257 KTRs used a low PPI dose, and 141

**Table 3. Subgroup analyses of the association of PPI use with mortality in 703 stable KTRs from the TransplantLines study.**

| | All-Cause Mortality | | | | | | |
|---|---|---|---|---|---|---|---|
| | No PPI | | Low PPI dose | | High PPI dose | | |
| Number of participants/events | 305/62 | | 257/80 | | 141/52 | | |
| | HR (95% CI) | P | HR (95% CI) | P | HR (95% CI) | P | $P_{trend}$ |
| Crude | Reference | n/a | 1.72 (1.23–2.39) | 0.001 | 2.14 (1.48–3.09) | <0.001 | <0.001 |
| Model 1 | Reference | n/a | 1.57 (1.10–2.23) | 0.01 | 2.03 (1.38–2.97) | <0.001 | <0.001 |
| Model 2 | Reference | n/a | 1.57 (1.08–2.24) | 0.02 | 1.88 (1.27–2.77) | 0.002 | 0.001 |
| Model 3 | Reference | n/a | 1.53 (1.05–2.22) | 0.03 | 1.90 (1.28–2.83) | 0.002 | 0.001 |
| Model 4 | Reference | n/a | 1.52 (1.06–2.19) | 0.05 | 1.81 (1.22–2.48) | 0.003 | 0.002 |
| Model 5 | Reference | n/a | 1.41 (0.97–2.04) | 0.07 | 1.62 (1.09–2.42) | 0.02 | 0.02 |
| Model 6 | Reference | n/a | 1.39 (0.97–1.99) | 0.1 | 1.57 (1.06–2.32) | 0.03 | 0.02 |
| Model 7 | Reference | n/a | 1.47 (1.02–2.12) | 0.04 | 1.62 (1.09–2.44) | 0.02 | 0.02 |

Model 1: PPI use adjusted for age, sex, BMI, and time since transplantation. Model 2: Model 1 additionally adjusted for eGFR, proteinuria, deceased donor transplant, preemptive transplantation, and primary renal disease. Model 3: Model 2 additionally adjusted for donor age, donor sex, donor weight, donor height, donor serum creatinine, number of HLA mismatches, and induction therapy. Model 4: Model 2 additionally adjusted for smoking behavior and alcohol use. Model 5: Model 2 additionally adjusted for the use of antihypertensive agents, platelet inhibitors, vitamin K antagonists, proliferation inhibitors, and CNIs. Model 6: Model 2 additionally adjusted for comorbidities (diabetes, history of cardiovascular disease). Model 7: Model 2 additionally adjusted for potential mediators (plasma magnesium and serum iron).

**Abbreviations:** BMI, body mass index; eGFR, estimated glomerular filtration rate; HLA, Human Leukocyte Antigen; HR, hazard ratio; KTR, kidney transplant recipient; n/a, not applicable; PPI, proton-pump inhibitor

KTRs used a high PPI dose. The association between PPI use and mortality risk appeared to be dose dependent, with the higest risk for premature mortality found among KTRs taking more than 20 mg omeprazole equivalents/day ($P_{trend} < 0.001$, Table 3).

## 3.2 Results replication study

Baseline characteristics from the Leuven Renal Transplant Cohort are shown in S5 Table. PPIs were used by 329 KTRs (50.2%). In this cohort, PPI users were significantly older (57 ± 12 years versus 54 ± 13 years, $P = 0.001$) and had higher triglyceride levels and HbA1c levels. More PPI users had a history of cardiovascular disease compared with nonusers (27.4% versus 19.0%, $P = 0.01$). Additionally, use of platelet inhibitors and prednisolone was more common among PPI users compared with nonusers.

Median follow-up of the Leuven Renal Transplant Cohort was 3.7 years. During this period, 97 (17.2%) KTRs died with a functioning graft. Out of the 329 KTRs who used PPIs at baseline, 65 (19.8%) died during follow-up, whereas out of the 327 KTRs who did not use PPIs at baseline, 32 (9.8%) died during follow-up (log-rank test $P < 0.001$, Fig 2). Prospective analysis showed that PPI users had a more than 2 times higher risk of mortality compared with nonusers (HR 2.47, 95% CI 1.61–3.78, $P < 0.001$, crude model, S6 Table). Further adjustment for potential confounders, including age, sex, time since transplantation, preemptive transplantation, deceased donor transplant, and primary renal disease, did not substantially affect this association (HR 1.75, 95% CI 1.12–2.73, $P = 0.01$). These results were similar to results obtained in the TransplantLines study.

## 4. Discussion

This study shows that PPI use is associated with an increased mortality risk in 2 large independent cohort studies of stable KTRs. Although significant baseline differences between PPI

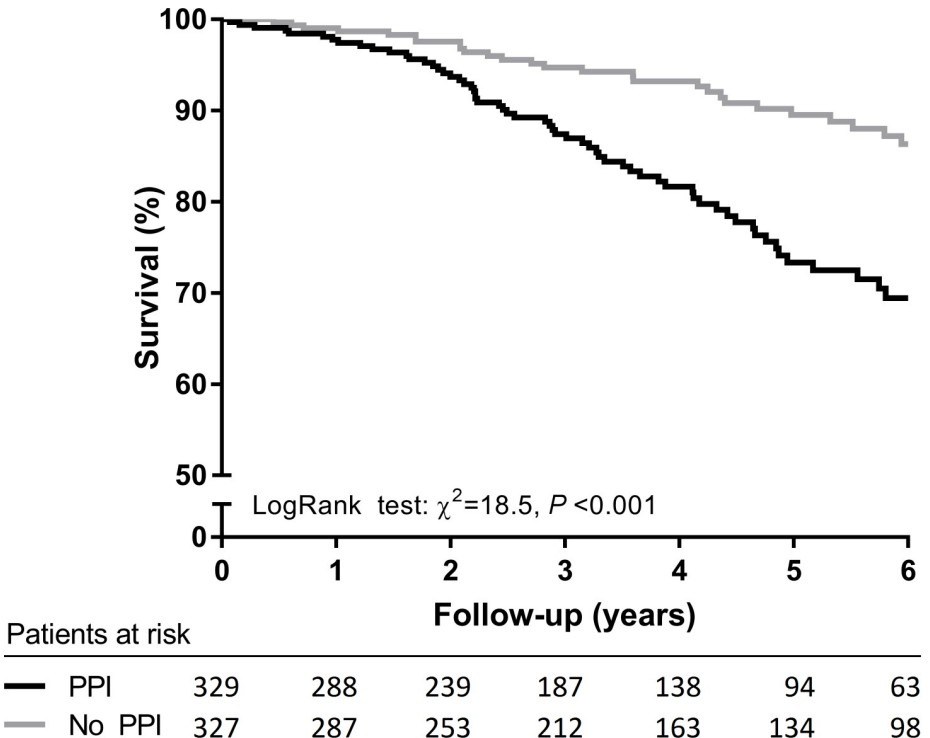

**Fig 2. Kaplan-Meier survival curve for all-cause mortality of PPI users compared with nonusers from the Leuven Renal Transplant Cohort.** PPI, proton-pump inhibitor.

users and KTRs who did not use PPIs were present, the association remained materially unchanged after adjustment for potential confounders. Moreover, we observed that the HR for mortality risk in KTRs taking a high PPI dose was higher than in KTRs taking a low PPI dose. Results from survival analysis in the Leuven Renal Transplant Cohort were similar to results obtained in the TransplantLines study.

Our main observation of an independent association between PPI use and increased mortality risk in KTR is in accordance with previous findings of a large ($n$ = 349,312) longitudinal observational cohort study among United States veterans. This study found an increased mortality risk among PPI users, compared with users of H2-receptor antagonists and participants who used neither PPIs nor H2-receptor antagonists [26]. Several small cohort studies among institutionalized elderly and older patients recently discharged from emergeny departments also demonstrated that PPI use is associated with an increased risk of mortality [23–25]. In addition, PPI use was found to be an independent predictor of mortality in 1,776 hemodialysis patients (HR 2.70, 95% CI 1.38–5.27, $P < 0.01$) [27]. In our study, the increased risk of premature mortality associated with PPI use was higher than previously reported by Xie and colleagues [26]. Apparently, the increase in risk arising from inhibition of gastric acid secretion is higher in KTRs than in the general population. It is conceivable that in light of active immunosuppression and an existing high burden of atherosclerosis, PPIs might increase susceptibility to serious infections and/or accelerate atherosclerosis, leading to more pronounced shortening of life expectancy.

Several physiological mechanisms may underlie the observed association between PPI use and mortality. It has been widely documented that PPIs may affect absorption of micronutrients, leading to deficiencies of important electrolytes, including iron and magnesium

[11,14,15]. Indeed, we previously found that PPI use was associated with iron deficiency and hypomagnesemia in KTRs [37,39]. Iron deficiency can in turn lead to iron deficiency anemia, which has been linked to a higher graft failure risk and mortality risk in KTRs [40–42]. Furthermore, low serum magnesium and urinary magnesium excretion are known risk factors for the development of hypertension, cardiovascular disease, and mortality in the general population [43–45], and low serum magnesium levels have been linked to mortality in patients with early stages of chronic kidney disease [46]. Additionally, PPI use has been associated with increased risk of cardiovascular mortality in the general population, which we also observed in our study [47]. It could be speculated that the higher mortality risk observed is attributable to lower iron and magnesium status in PPI users. However, when we adjusted for iron and magnesium levels in prospective analyses, the relationship between PPI use and mortality remained unaltered, implying that the observed risk associated with PPI use could not be attributed to lower iron and magnesium status and that other mechanisms are likely involved.

Another explanation might be that other adverse effects related to PPI use, such as an increased risk of gastrointestinal infections, community-acquired pneumonia, and acute and chronic kidney disease, collectively carry an increased mortality risk [18–20,48,49]. Unfortunately, data on gastrointestinal infections and community-acquired pneumonia were unavailable in our study. Therefore, we were unable to investigate this hypothesis.

A relatively unexplored field in nephrology is the gut microbiome and its role in development of disease and adverse health outcome after renal transplantation. It has previously been shown that PPIs have the ability to drastically change the composition of gut microbiota resulting in a less healthy gut microbiota with a lower diversity and a tendency towards *C. difficile* and other enteric infections [50,51]. Moreover, gut microbial dysbiosis has been linked to post-transplantation complications such as rejection and graft versus host disease in allogeneic transplantation [52–54], demonstrating how important the gut microbiome might be in relation to adverse outcome after transplantation. Furthermore, Evenepoel and colleagues previously demonstrated that PPIs impair protein digestion, which results in higher protein availability in the colon and consequently in higher systemic levels of potential nephrotoxic microbial fermentation products such as p-cresol [55]. However, future research is essential to elucidate the interplay between PPI use, alterations in the gut microbiome, and mortality after renal transplantation.

Further prospective analysis showed that PPI use was not associated with a higher risk of graft failure. These findings are consistent with findings from Knorr and colleagues, who did not find an association between PPI use and graft failure in a cohort of 597 KTRs [56]. In the present study, a majority of patients (56.6%) used PPIs, indicating the frequent use of PPIs among KTRs. Chronic use of PPIs has tremendously increased over the past decade, and studies estimate that in both primary and hospital care 30% to 65% of patients that chronically use PPIs are using it for an inappropriate indication [57–61], including, e.g., corticosteroid therapy without concomitant nonsteroidal anti-inflammatory drug (NSAID) use [61,62]. Inappropriate use of PPIs may be frequent in KTRs, since PPIs are commonly prescribed to prevent gastrointestinal complaints and complications of immunosuppressive medication, particularly of corticosteroid therapy [63]. According to the Food and Drug Administration (FDA) guidelines, PPI use is not routinely indicated in this situation [64,65].

Our results might have important implications for clinical practice. The present study highlights the importance of an evidence-based indication for PPI treatment and suggests that treatment indication may need to be revisited in KTRs. Hence, physicans should deliberate whether the benefits of PPI therapy outweigh the risks for each individual patient. Moreover, rebound acid hypersecretion, a phenomenon that can occur after PPI cessation, might complicate treatment discontinuation [66,67]. It is therefore important that physicians are aware of

this phenomenon and inform patients about this potential rebound effect while withdrawing PPI treatment.

One of the strenghts of this study is the use of a prospectively followed cohort of well-characterized KTRs, in which endpoint evaluation was complete with no loss to follow-up. The fact that we used data from a cohort with extensivley phenotyped participants enabled us to correct for many possible confounding factors, including lifestyle factors, medication use, and comorbidities. Moreover, independent replication in the Leuven Renal Transplant Cohort showed similar results, which strengthens the evidence for an association between PPI use and mortality risk in KTR.

However, some limitations need to be taken into consideration. First, the fact that participants were predominantly Western European Caucasian limits the generalizability of the results to other populations. In addition, results may not be generalizable to KTRs with opportunistic infections and patients with overt heart failure given that these patients were not included in the study. Second, due to the observational design of both studies, a cause-effect relationship cannot be established with certainty, and despite adjustment for various potential confounders, the possibility of residual confounding due to unknown or unmeasured variables remains. On average, PPI users had more risk factors for mortality than nonusers, with the result that the contribution of PPIs may have been overestimated. Further prospective investigation is needed to validate whether chronic PPI use leads to increased mortality in KTRs or whether KTRs with increased mortality risk are subjected to more frequent treatment with PPIs. Furthermore, PPI users were included more shortly after transplantation, which may have resulted in a survival selection bias. However, prospective analyses with data from the replication cohort showed similar results. In this cohort, time after transplantation was not significantly different between PPI users and nonusers. In addition, data on donor-specific antigens were not available in our cohort, and we could therefore not adjust for this potential confounder. Lastly, indications for PPI use such as peptic ulcer disease and gastroesophageal reflux disease may increase risk of death due to malignant diseases. Although confounding by indication becomes less likely with results of cause-specific analyses, which did not demonstrate an increased mortality risk from malignant diseases, it cannot be excluded.

In conclusion, we demonstrated that PPI use is associated with an increased risk of all-cause mortality, with the highest risk among PPI users exposed to a high PPI dose. Further research is necessary to reveal the mechanism by which PPI use increases mortality risk in KTRs.

## Supporting information

**S1 STROBE Checklist.**
(DOCX)

**S1 TRIPOD Checklist.**
(DOCX)

**S1 Appendix.**
(DOCX)

**S1 Table. Association of PPI use with mortality in 703 stable KTRs.** Results from analyses in the nonimputed dataset from the TransplantLines study. Model 1: PPI use adjusted for age, sex, BMI, time since transplantation. Model 2: Model 1 additionally adjusted for eGFR, proteinuria, deceased donor transplant, preemptive transplantation, primary renal disease. Model 3: Model 2 additionally adjusted for donor age, donor sex, donor weight, donor height, donor serum creatinine, number of HLA mismatches, and induction therapy. Model 4: Model

2 additionally adjusted for smoking behavior and alcohol use. Model 5: Model 2 additionally adjusted for use of antihypertensive agents, platelet inhibitors, vitamin K antagonists, proliferation inhibitors, and CNIs. Model 6: Model 2 additionally adjusted for comorbidities (diabetes, history of cardiovascular disease). Model 7: Model 2 additionally adjusted for potential mediators (plasma magnesium and serum iron).
(DOCX)

**S2 Table. Association of PPI use with cause-specific mortality in 703 stable KTRs.** Model 1: PPI use adjusted for age, sex, time since transplantation. Model 2: Model 1 additionally adjusted for eGFR, deceased donor transplant, preemptive transplantation, primary renal disease.
(DOCX)

**S3 Table. Association of PPI use with graft failure in 703 stable KTRs.** Model 1: PPI use adjusted for age, sex, time since transplantation. Model 2: Model 1 additionally adjusted for eGFR, deceased donor transplant, preemptive transplantation, primary renal disease.
(DOCX)

**S4 Table. Association between PPI use and change in renal function during follow-up.** Model 1: PPI use adjusted for time from baseline until follow-up. Model 2: Model 1 additionaly adjusted for age, sex, and BMI.
(DOCX)

**S5 Table. Baseline characteristics of 656 KTRs from the Leuven Renal Transplant Cohort.** Data are presented as mean ± SD, median with IQRs, or number with percentages (%). [a]Missing in 354 cases; [b]missing in 299 cases. BMI, body mass index; eGFR, estimated glomerular filtration rate; HbA1c, hemoglobin A1c; HDL, high-density lipoprotein; IQR, interquartile range; LDL, low-density lipoprotein.
(DOCX)

**S6 Table. Association of PPI use with mortality in 656 stable KTRs from the Leuven Renal Transplant Cohort.** Model 1: PPI use adjusted for age, sex, time since transplantation. Model 2: Model 1 additionally adjusted for eGFR, deceased donor transplant, preemptive transplantation, primary renal disease.
(DOCX)

## Author Contributions

**Conceptualization:** Rianne M. Douwes, António W. Gomes-Neto, Michele F. Eisenga, Elisabet Van Loon, Joëlle C. Schutten, Rijk O. B. Gans, Maarten Naesens, Else van den Berg, Ben Sprangers, Stefan P. Berger, Gerjan Navis, Hans Blokzijl, Björn Meijers, Stephan J. L. Bakker, Dirk Kuypers.

**Data curation:** Rianne M. Douwes, Elisabet Van Loon, Maarten Naesens, Else van den Berg, Björn Meijers.

**Formal analysis:** Rianne M. Douwes, António W. Gomes-Neto.

**Funding acquisition:** Else van den Berg, Björn Meijers, Stephan J. L. Bakker, Dirk Kuypers.

**Methodology:** Rianne M. Douwes, António W. Gomes-Neto, Michele F. Eisenga, Elisabet Van Loon, Joëlle C. Schutten, Rijk O. B. Gans, Maarten Naesens, Else van den Berg, Ben Sprangers, Stefan P. Berger, Gerjan Navis, Hans Blokzijl, Björn Meijers, Stephan J. L. Bakker, Dirk Kuypers.

**Project administration:** Else van den Berg.

**Supervision:** Stefan P. Berger, Gerjan Navis, Hans Blokzijl, Björn Meijers, Stephan J. L. Bakker, Dirk Kuypers.

**Writing – original draft:** Rianne M. Douwes, António W. Gomes-Neto.

**Writing – review & editing:** Rianne M. Douwes, António W. Gomes-Neto, Michele F. Eisenga, Elisabet Van Loon, Joëlle C. Schutten, Rijk O. B. Gans, Maarten Naesens, Else van den Berg, Ben Sprangers, Stefan P. Berger, Gerjan Navis, Hans Blokzijl, Björn Meijers, Stephan J. L. Bakker, Dirk Kuypers.

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
