## [Editor Report · Decision Letter 0]

21 Jan 2020

Dear Dr Douwes, 

Thank you for submitting your manuscript entitled "Use of Proton-Pump Inhibitors and Excess Mortality after Kidney Transplantation: A Prospective Cohort Study with Independent Replication." for consideration by PLOS Medicine.

Your manuscript has now been evaluated by the PLOS Medicine editorial staff and I am writing to let you know that we would like to send your submission out for external peer review.

Kind regards,

Helen Howard, for Clare Stone PhD 

Acting Editor-in-Chief

PLOS Medicine 

plosmedicine.org

---

## [Decision Letter · Decision Letter 1]

28 Jan 2020

Dear Dr. Douwes,

Thank you very much for submitting your manuscript "Use of Proton-Pump Inhibitors and Excess Mortality after Kidney Transplantation: A Prospective Cohort Study with Independent Replication." (PMEDICINE-D-20-00132R1) for consideration at PLOS Medicine. 

[LINK]

In light of these reviews, I am afraid that we will not be able to accept the manuscript for publication in the journal in its current form, but we would like to consider a revised version that addresses the reviewers' and editors' comments. Obviously we cannot make any decision about publication until we have seen the revised manuscript and your response, and we plan to seek re-review by one or more of the reviewers. 

We expect to receive your revised manuscript by Feb 11 2020 11:59PM. Please email us (plosmedicine@plos.org) if you have any questions or concerns.

We look forward to receiving your revised manuscript. 

Sincerely,

Adya Misra, PhD

Senior Editor 

PLOS Medicine

plosmedicine.org

Title: please revise to adhere to PLOS Medicine style; the colon should be followed by study descriptor only. We suggest you revise to “The association between use of Proton-Pump Inhibitors and Excess Mortality after Kidney Transplantation: A Prospective Cohort Study” 

Abstract- last sentence of the methods and findings section should be a limitation of your methodology 

Data availability- as per PLOS data policy, authors cannot be direct contacts for data requests. Please provide third party access, such as a data or ethics committee who can be contacted by interested authors. 

Author summary-At this stage, we ask that you include a short, non-technical Author Summary of your research to make findings accessible to a wide audience that includes both scientists and non-scientists. The Author Summary should immediately follow the Abstract in your revised manuscript. This text is subject to editorial change and should be distinct from the scientific abstract. Please see our author guidelines for more information: https://journals.plos.org/plosmedicine/s/revising-your-manuscript#loc-author-summary

Throughout- please move the full stop before references to after, with a space in between text and brackets but not brackets and full stop. Please also provide square brackets for references. For example “ xxx [1].” 

Methods Line 136 – please provide details of the questionnaire used, citing it if previously published or providing a copy as supplementary information

Discussion- please avoid assertions of primacy 

Line 344-345 should contain “ a cause-effect relationship” 

Did your study have a prospective protocol or analysis plan? Please state this (either way) early in the Methods section.

c) In either case, changes in the analysis—including those made in response to peer review comments—should be identified as such in the Methods section of the paper, with rationale.

Please ensure that the study is reported according to the STROBE guideline, and include the completed checklist as Supporting Information. When completing the checklist, please use section and paragraph numbers, rather than page numbers. Please add the following statement, or similar, to the Methods: "This study is reported as per the Strengthening the Reporting of Observational Studies in Epidemiology (STROBE) guideline (S1 Checklist)."

Please report your study according to the relevant guideline, which can be found here: http://www.equator-network.org/

 transparent reporting of a multivariable prediction model for individual prognosis or diagnosis (TRIPOD) guideline (S1 Checklist).

Comments from the reviewers:

Reviewer #1: I liked this paper. Very well thought out . well controlled for distracting variables. We wrote on this.... "Concomitant Proton Pump Inhibitors with Mycophenolate Mofetil and the Risk of Rejection in Kidney Transplant Recipients. J. Knorr, M. Sjeime, L. Braitman, P. Jawa, R. Zaki, J. Ortiz. Transplantation March 15 2014. Vol 97 issue 5". Your team did not use.

I like it. well written. good conclusion.

Reviewer #2: The investigators conducted this study to evaluate whether PPI use is associated with mortality risk in kidney transplant recipients.

While this is important study, the findings of this study alone to evaluate mortality outcome might not adequately worth publishing in this high impact journal without significant modifications/revisions. There are many factors that can affect mortality risk in kidney transplant patients, such as cardiovascular, infection, rejection, allograft outcomes. However, the investigators barely look into this.

It is possible that the investigators plan to separate these outcomes to several publications, since the investigators have published their works by using this cohort to evaluate several outcomes e.g. hypomagnesemia (PMID: 31817776) and iron status (PMID: 31484461)

Without knowing the causes of mortality and lack of granularity (in addition to just 'mortality risk"), this finding of this study does not add much into the field of transplantation.

Reviewer #3: I applaud the authors on this significant work. This is an important study as it adds to the growing literature supporting an association between PPI and increased mortality, specifically involving kidney transplant recipients. The authors have provided two different cohorts demonstrating an association between PPI use and increased mortality. Based on current knowledge, I would not be surprised to find that such a positive correlation does exist between PPI and mortality. However, I question whether the reported risk is as great as presented as noted below.

Page 7, line 107. Looking at their prior described cohort (reference 28), this study appears to be a retrospective rather than prospective study as the prior cohort was established with an aim that would not have accrued the same information as presented in this study. If the authors can confirm this is accurate, the manuscript should be adjusted accordingly. 

Page 7, line 112. What other KTR patients were excluded as noted for "concurrent systemic illnesses" aside from the named malignancies other than cured skin cancer, opportunistic infections, and overt congestive heart failure? 

It should be noted that this study may not be generalizable to KTR patients who had these systemic illnesses. As opportunistic infections can be an acute process rather than chronic, it would have been helpful to include these patients. In addition, it would be interesting to know the results from patients with overt CHF as these patients may have an increased atherosclerotic burden compared to non-CHF patients. As such, these patients 

The patients in the non-PPI group had a significantly increased time since transplantation compared to the PPI group. This would seem to cause a significant survival selection bias. This may be due to the retrospective nature (see above regarding question of retrospective vs prospective) of this study. If this was a prospective study as the authors stated, then performing a study where patients were at the same timepoint after kidney transplant with or without PPI would have helped avoid this significant survival bias (e.g. picking recipients who received their kidney transplant a certain year and then evaluating their mortality outcomes based on PPI vs non-PPI). I appreciate that the authors adjusted for this factor, but as we know, there are many unknown confounders that may be present in an individual who has survived with a functional graft already for 9 years vs an individual for 4 years.

Table 1: It would have been interesting to include antiplatelet use. Previously, there was some concern that PPI's may interfere with clopidogrel efficacy, and KTR are at high risk for CV complications and mortality. Furthermore, since patients on dual antiplatelet medications for coronary atherosclerosis either before or after stenting tend to also be on PPI's, these patients are likely at higher risk for CV complications and mortality. The omission of patients with overt heart failure and not adjusting for antiplatelet agents may influence the results of this study.

Page 11, line 220. What percentage of patients from the PPI vs non-PPI group developed graft failure? Along this line, was the eGFR was only determined at the initial baseline measurement or was it determined closer to patient death? Since tubulointerstitial nephritis (acute or chronic) is associated with PPI's, it would have been useful to see if PPI patients also had a decreased eGFR at time of death. Since renal function is associated with mortality, this would be another important covariate to control for (aside from what the authors had already done by omitting patients who developed graft failure.

Table 2 and Table 3: Was there a model where you adjusted for all of the models (1-6) combined? What was this result?

I appreciate the authors providing several mechanisms for PPI use and increased mortality. 

Minor english revisions recommended to help improve readability.

Reviewer #4: The authors have attempted to investigate the effect of PPI therapy on mortality in a large single center cohort of stable outpatient KTR, and performed independent replication of the results in a second cohort. 

Cohort demographics and characteristics - is the sample representative of wider population? It is noted from the discussion on limitations that subjects were predominantly Caucasian - what other comparisons can be made to better understand generalisability of the results? It is also noted that the data used is from a previously described cohort, but a brief overview or summary within this manuscript would be helpful to the reader in their assessment of how robust the findings and conclusions are. 

Confounding has been accounted for adequately, given the limitations of the data, and the description, production and presentation of all 6 models is very useful. 

Under methods:

"Follow-up was recorded until September 2015. "

Can follow-up now be extended? The current data has median follow-up of 5.3 [4.5-6.0] years, which could almost be doubled in length adding great value to the longitudinal aspects of this study.

The statistical techniques appear to be appropriate for the data and the research question in hand.

It is also good to see that a sensitivity analysis of missing data has been performed, by comparing mulitple imputation results to no imputation.

Did the authors undertake any sub-group anaylses (by which, I am not referring to the dose-response analysis shown in Table 3)? 

Was there an association between outcome and age, for instance, as suggested by the literature [23-25]?

The conclusions are fair given the study design undertaken, and the manuscript as a whole is well written, clear and concise.

The Tables and Figures are also clear and informative.

[LINK]

---

## [Decision Letter · Decision Letter 2]

22 Apr 2020

Dear Dr. Douwes,

Thank you very much for re-submitting your manuscript "The Association between Use of Proton-Pump Inhibitors and Excess Mortality after Kidney Transplantation: A Prospective Cohort Study." (PMEDICINE-D-20-00132R2) for review by PLOS Medicine.

I have discussed the paper with my colleagues and the academic editor and it was also seen again by xxx reviewers. I am pleased to say that provided the remaining editorial and production issues are dealt with we are planning to accept the paper for publication in the journal.

[LINK]

We look forward to receiving the revised manuscript by Apr 29 2020 11:59PM. 

Sincerely,

Adya Misra, PhD

Senior Editor 

PLOS Medicine

plosmedicine.org

Requests from Editors:

Title- please remove the word “prospective” since the analyses were carried out after the cohort had already been established

Author summary

Line 82-please revise to “in the long term”

Introduction

Line 117- could you rephrase “gastrointestinal complaints” for specificity? 

Methods

Please provide the information about your replication cohort earlier in the methods section within 2.1- Design and Study population. Please also remove lines 265-269 as these are better placed in introduction/discussion sections

Results

Line 300-301 do you mean “excluded from analyses” instead of “censored”? 

In the Kaplan-Meier curve in Figures 1 and 2, please provide the number at risk for each time interval.

Comments from Reviewers:

Reviewer #2: -My major concerns still persist. Patients who received PPI usually are different than those who received PPI. Those on PPI usually have high cardiovascular disease and on antiplatelet agents and/or anticoagulations,

Medications such as aspirin, other antiplatelets/anticoagulation should be taken into consideration.

-Also, more data on transplant-related are needed. Data on induction therapy, and HLA-mismatch, and DSA, and KDPI, which are important data for kidney transplant studies, have not been included and adjusted. 

-Causes of death have not been look at. PPI per se does not cause mortality, but the differences in characteristics, commodities, that cause mortality. Rejection as outcome also has not been demonstrated in this study.

Reviewer #4: The authors have responded to each comment in turn, adding data, completing further analyses, and amending the manscript contents accordingly.

[LINK]

---

## [Editor Report · Decision Letter 3]

13 May 2020

Dear Dr. Douwes, 

On behalf of my colleagues and the academic editor, Dr. Maarten Taal, I am delighted to inform you that your manuscript entitled "The Association between Use of Proton-Pump Inhibitors and Excess Mortality after Kidney Transplantation: A Cohort Study." (PMEDICINE-D-20-00132R3) has been accepted for publication in PLOS Medicine. 

PRODUCTION PROCESS

PRESS

PROFILE INFORMATION

Thank you again for submitting the manuscript to PLOS Medicine. We look forward to publishing it. 

Best wishes, 

Adya Misra, PhD

Senior Editor 

PLOS Medicine

plosmedicine.org